# Performance of IOTA Simple Rules Risks, ADNEX Model, Subjective Assessment Compared to CA125 and HE4 with ROMA Algorithm in Discriminating between Benign, Borderline and Stage I Malignant Adnexal Lesions

**DOI:** 10.3390/diagnostics13050885

**Published:** 2023-02-25

**Authors:** Artur Czekierdowski, Norbert Stachowicz, Agata Smolen, Tomasz Łoziński, Paweł Guzik, Tomasz Kluz

**Affiliations:** 1Department of Gynecological Oncology and Gynecology, Medical University of Lublin, 20-081 Lublin, Poland; 2Chair and Department of Epidemiology and Clinical Research Methodology, Medical University of Lublin, 20-080 Lublin, Poland; 3Department of Obstetrics and Gynecology, Pro-Familia Hospital, 35-302 Rzeszow, Poland; 4Department of Obstetrics and Gynecology, Fryderyk Chopin University Hospital No. 1, Faculty of Medicine, Rzeszow University, 35-310 Rzeszow, Poland; 5Department of Gynecology and Obstetrics, City Hospital, 35-241 Rzeszow, Poland

**Keywords:** complex adnexal mass, ovarian cancer, borderline ovarian tumors, HE4, CA125, risk of malignancy algorithm (ROMA), International Ovarian Tumor Analysis (IOTA), Simple Rules Risk, ADNEX model

## Abstract

BACKGROUND: Borderline ovarian tumors (BOTs) and early clinical stage malignant adnexal masses can make sonographic diagnosis challenging, while the clinical utility of tumor markers, e.g., CA125 and HE4, or the ROMA algorithm, remains controversial in such cases. OBJECTIVE: To compare the IOTA group Simple Rules Risk (SRR), the ADNEX model and the subjective assessment (SA) with serum CA125, HE4 and the ROMA algorithm in the preoperative discrimination between benign tumors, BOTs and stage I malignant ovarian lesions (MOLs). METHODS: A multicenter retrospective study was conducted with lesions classified prospectively using subjective assessment and tumor markers with the ROMA. The SRR assessment and ADNEX risk estimation were applied retrospectively. The sensitivity, specificity, positive and negative likelihood ratios (LR+ and LR−) were calculated for all tests. RESULTS: In total, 108 patients (the median age: 48 yrs, 44 postmenopausal) with 62 (79.6%) benign masses, 26 (24.1%) BOTs and 20 (18.5%) stage I MOLs were included. When comparing benign masses with combined BOTs and stage I MOLs, SA correctly identified 76% of benign masses, 69% of BOTs and 80% of stage I MOLs. Significant differences were found for the presence and size of the largest solid component (*p* = 0.0006), the number of papillary projections (*p* = 0.01), papillation contour (*p* = 0.008) and IOTA color score (*p* = 0.0009). The SRR and ADNEX models were characterized by the highest sensitivity (80% and 70%, respectively), whereas the highest specificity was found for SA (94%). The corresponding likelihood ratios were as follows: LR+ = 3.59 and LR− = 0.43 for the ADNEX; LR+ = 6.40 and LR− = 0.63 for SA and LR+ = 1.85 with LR− = 0.35 for the SRR. The sensitivity and specificity of the ROMA test were 50% and 85%, respectively, with LR+ = 3.44 and LR− = 0.58. Of all the tests, the ADNEX model had the highest diagnostic accuracy of 76%. CONCLUSIONS: This study demonstrates the limited value of diagnostics based on CA125 and HE4 serum tumor markers and the ROMA algorithm as independent modalities for the detection of BOTs and early stage adnexal malignant tumors in women. SA and IOTA methods based on ultrasound examination may present superior value over tumor marker assessment.

## 1. Introduction

Malignant ovarian masses and, especially, ovarian cancers are marked by exceptionally high mortality rates, mostly due to the lack of characteristic symptoms and the late diagnosis [1]. Despite ongoing advances in surgical techniques and in pharmacological approaches, the early detection of malignancy is the only way to achieve a better cure rate in affected women [2]. Although imaging methods and selected tumor marker assessment assist in the detection and staging of invasive lesions, their efficacy is limited in borderline tumors and early stage ovarian cancers, and specific diagnosis is hampered by the lack of biopsy specimens [3]. Moreover, ovarian cyst fluid cytology, although highly specific, was only moderately sensitive for the detection of ovarian malignancies [4]. 

Accurate preoperative differentiation between benign masses, borderline ovarian lesions and malignant ovarian masses may impact surgical management. Minimally invasive surgery, such as laparoscopy with an endobag or minilaparotomy, is recommended for borderline tumors and selected early stage ovarian cancers if there is no risk of tumor rupture [5]. Fertility-sparing surgery (FSS) is also a strategy commonly considered in young patients with early epithelial ovarian cancer [6]. Evaluating the presence of possible malignant disease in women with pelvic adnexal masses currently relies on medical imaging and serum marker findings [7]. 

In theory, accurate biomarkers that aid risk stratification could improve the current diagnostic and decision-making quandary in patients with adnexal masses. Tumor markers, such as CA125 and HE4, or their combination known as the ROMA test only have a limited role in the preoperative discrimination of adnexal mass type [8,9,10]. The use of CA125 sensitivity and HE4 specificity along with the patient’s age and her menopausal status could be superior to tumor markers used alone [11]. More recent studies have questioned the practical clinical value of the ROMA test for diagnosing ovarian masses, especially when compared to MRI and ultrasound-based discrimination systems [12,13,14]. The initial diagnosis made via pelvic sonography strongly depends on the operator’s experience, which, in turn, may lead to inconsistent results [15,16]. Several scoring systems, such as the IOTA group Simple Rules, the LR1 and LR2 IOTA logistic regression models and the Assessment of Different NEoplasias in the adneXa (ADNEX model), are recommended for less experienced operators [17]. The use of the Simple Rules (SRs) was reported to have a sensitivity of 93% and a specificity of 90% and showed the potential to improve the management of women with adnexal masses [18]. However, as many as 25% of adnexal masses may not be classified with that method. According to various studies, the rate of malignancy in the indeterminate category of adnexal cysts may vary from 5 to 40%, contributing to a risk for either a delay in treatment if followed conservatively or a referral for an unnecessary surgery [19,20]. In order to improve the predictive value, the IOTA group proposed a modification of the SR method called the Simple Rules Risk (SRR) [21]. This logistic regression model allows the calculation of the probability of malignancy in almost all types of adnexal masses and does not use CA125 for risk estimation. Another IOTA group logistic regression model called the ADNEX was validated in multiple external studies and showed good to excellent predictive values [22]. More recently, Hiett et al. found that the IOTA group proposed methods, i.e., the SR, SRR and ADNEX models and the O-RADS classification, were characterized by comparable sensitivity in the preoperative discrimination of malignant from benign pelvic tumors [23]. However, despite the use of subjective assessment by an expert examiner, approximately 7–8% of adnexal masses that are considered appropriate for surgical removal may not be correctly classified as either benign or malignant due to their complex morphology and/or small size [24,25]. 

The purpose of this study was to apply the IOTA group Simple Rules Risk (SRR) estimation, the ADNEX model and the subjective assessment (SA) in the same cohort of patients with complex adnexal masses and to compare their performance with serum CA125 and HE4, along with the ROMA test, in the preoperative discrimination between benign adnexal masses, borderline tumors and stage I malignant ovarian lesions.

## 2. Materials and Methods

This was a multicenter retrospective study that included women who were operated on because of complex morphology adnexal tumors. The study was carried out in two departments of obstetrics and gynecology (Department of Obstetrics and Gynecology at the Fryderyk Chopin University Hospital No 1 and Department of Obstetrics and Gynecology of the ProFamilia Hospital, Rzeszow, Poland) and one Gynecological Oncology Center (Department of Gynecological Oncology and Gynecology of the Medical University of Lublin, Poland) between January 2012 and December 2020. High-end ultrasound equipment was used for the purpose of this study: GE Voluson E8 and E10 (GE Healthcare, Zipf, Austria) or Samsung WS80A Elite and Hera W10 (Samsung, Seoul, Korea). The frequencies of the vaginal probes varied between 5.0 and 9.0 MHz and those of the abdominal probes between 3.5 and 5.0 MHz. Patients were eligible for this study if they were aged 18 years or older at recruitment and had at least 1 complex morphology adnexal mass (ovarian, paraovarian or tubal) detected on ultrasonographic examination. Complex adnexal mass was defined as a non-unilocular cyst or a cystic–solid lesion with non-uniform echogenicity (Figure 1). 

Other inclusion criteria were as follows: a known expression of CA125 and HE4 serum tumor markers measured preoperatively and tumor microscopic examination available, confirming either benign (BEN) or borderline ovarian tumor (BOT) or early stage (FIGO stage I) malignant ovarian lesion (MOL) according to 2014 WHO classification of tumors of the female reproductive organs [26]. The serum levels of CA125 and HE4 were measured with the use of chemiluminescent enzyme immunoassay with the Cobas Integra, CobasE411 and Cobas 6000 (Roche, Switzerland) analyzers. A cut-off value of 35 U/mL was used for CA125 and for HE4. The corresponding cut-off levels of 70 pmol/L for premenopausal women and 105 pmol/L for postmenopausal patients were used, respectively. 

A standardized examination protocol of transvaginal ultrasonographic examination (supplemented with a transabdominal scan, if necessary) was performed by level II or level III examiners in all cases [27,28]. Using the IOTA terminology, we searched for pelvic tumor features that were, or could have been, related to the increased risk of malignancy. Such features included tumor type (unilocular, multilocular, unilocular-solid, multilocular-solid and solid), cyst echogenicity (anechoic, hypoechoic, ground-glass echogenicity, hyperechoic or not applicable in solid lesions), tumor size (<40 mm, 40–100 mm, >100 mm), an ovarian crescent sign and acoustic shadows. When papillary projections equal to or higher than 3 mm were found within the cystic mass, their number, vascularity, contour and the presence of the microcystic pattern were also evaluated [28]. If present, the largest solid component other than a papillary projection was also measured (Figure 2). 

Tumor vascularity was assessed with a color score proposed by the IOTA group. A color score of 1 denotes that no color or power Doppler signal was found in the tumor, a score of 2 means that a minimal amount of color Doppler signals were detected, a color score of 3 means a moderate amount found and a score of 4 translates into abundant color Doppler signals being detected [28]. Figure 3 shows an example of IOTA color score 3 indicating a moderate vascularity in a solid adnexal mass.

### 2.1. Prognostic Models

The performance of six prognostic methods: CA125, HE4 and the ROMA, along with subjective assessment (SA) and the IOTA group SRR and ADNEX, was evaluated. On ultrasound examination, all adnexal masses were classified as probably benign or probably malignant according to the subjective assessment. The Simple Rules Risk (SRR) and IOTA ADNEX models were applied to obtain a risk score for each studied case. The SRR enables the calculation of a predicted probability of ovarian malignancy. We used the following 3 categories of SRR risk for statistical purposes: low (<3%), intermediate (3–20%) and high (>20%). The ADNEX model provides the probability of a benign mass and the predicted risks of four different subclasses of malignant adnexal tumors (borderline, stage I invasive, stage II–IV invasive or metastatic cancer). The index values of both IOTA models were computed according to the published algorithms [29,30]. A cut-off of 20% for the ADNEX-calculated probability of malignancy was used to identify women with a suspected malignant lesion as suggested by the original IOTA group study [31]. As regards the ROMA test, separate risk cut-offs for premenopausal (score > 1.14) and postmenopausal patients (score > 2.99) were applied, according to the test manufacturer’s instructions [32].

### 2.2. Statistical Methods

All the tests were assessed in terms of their ability to discriminate between 3 groups of patients. Group A comprised women with benign adnexal masses (BEN), group B included women with borderline tumors (BOTs) and group C included patients diagnosed with malignant ovarian lesions (MOLs). Categorical variables were reported as absolute frequencies (n) and proportions (%), and continuous variables as the mean ± standard deviation or median (interquartile range). The sensitivity, specificity, accuracy, false positive and false negative rates, as well as positive and negative likelihood ratios (LR+ and LR−), were calculated for all the tests. Areas under ROC curves (AUROCs) were calculated for CA125 and HE4 tumor markers, the ROMA test and for both IOTA group prognostic models. The accuracy, sensitivity, specificity, positive predictive value (PPV) and negative predictive value (NPV) and their 95% confidence intervals (CI) were also calculated to evaluate the diagnostic performance of CA125, HE4, the ROMA, and the SRR at recommended cut-off points. Statistical analysis was performed using STATISTICA v.13 (Statsoft, Tulsa, OK, USA). The level of significance was set at two-sided 5% (i.e., 0.05).

## 3. Results

The median age of the studied women (n = 108) was 48 years, and 44 of them were postmenopausal. The final histology confirmed 62 (57.4%) benign masses, 26 (24.1%) BOTs and 20 (18.5%) stage I MOLs. The histological diagnosis of all masses is presented in Table 1. 

Figure 4 shows grayscale and color Doppler images of a small ovarian round lesion in a 34-year-old patient. The maximum diameter of the lesion was 30 mm, and because of a uniform hypoechogenic appearance, it was originally thought to be an endometrioma (Figure 4a). However, color Doppler sonography revealed abundant central vascularization within the hypoechogenic area and allowed the correct reclassification of the lesion as a solid one (Figure 4b).

Figure 5 presents a sonographic color Doppler image of a small cystic–solid ovarian lesion that was preoperatively classified by subjective assessment as a borderline serous tumor due to the presence of an irregular contour of the vascularized papillary projection. The final histology revealed early stage invasive serous ovarian cancer.

Table 2 presents data on patients’ age with the menopausal status and selected ultrasonographic features of the studied adnexal masses. The results are shown as medians (interquartile range) for continuous and ordinal variables, and as *N* (%) for categorical variables.

Women with borderline tumors and patients with malignant adnexal masses were older than women with benign tumors. A substantial fraction (23%) of borderline tumors were smaller than 40 mm. The corresponding proportions for benign and malignant invasive tumors were 6% and 5%, respectively. Of the three studied groups, women with borderline adnexal masses also had the highest proportion of large, i.e., >100 mm, tumors. None of the malignant or borderline adnexal lesions were unilocular cysts.

Table 3 presents the comparisons of selected sonographic tumor features in the studied group of women with complex adnexal masses.

Significant differences between the groups of benign and borderline/malignant invasive lesions were found for the presence and size of the largest solid component (*p* = 0.0006), the number of papillary projections (*p* = 0.01), papillation contour (*p* = 0.008) and IOTA color score (*p* = 0.0009). Based on the presented tumor morphologic features, subjective assessment by an expert examiner correctly identified 76% of benign masses, 69% of BOTs and 80% of stage I MOLs. Table 4 presents the preoperative serum concentrations of two tumor markers, CA125 and HE4, in the studied group of women.

Regarding the predictive values of the expression of both tumor markers, no significant differences between the studied groups were found. Table 5 presents data on the risk stratification of the studied groups with the prognostic algorithms used in this study.

The stratification of the ROMA and SRR test results into the classes of high and low risk indicated that both methods provided good differentiation between the various types of adnexal masses. The lowest values of index tests, the ROMA and SRR, were found in women with benign masses, with the differences between this group and both borderline and early stage malignancies being highly significant (*p* < 0.001). The ADNEX model had the highest predictive values in women with borderline tumors, and the differences between this group and both benign and malignant masses were also highly significant (*p* < 0.001). Subjective assessment enabled a correct diagnosis in 63% of cases in the whole group of women with complex adnexal masses. The differences between the three groups were highly significant as well (*p* < 0.001). Table 6 shows comparisons of the predictive values of predictive methods used in this study in the group of 62 women with benign masses and in 46 patients with borderline tumors or early stage malignant adnexal lesions. 

Regarding the IOTA group predictive models, the SRR and ADNEX correctly classified a vast proportion of adnexal masses as either benign or malignant. Both models had the highest sensitivity (80% and 70%, respectively), whereas the highest specificity was found for SA (94%). The corresponding likelihood ratios were as follows: LR+ = 3.59 and LR− = 0.43 for the ADNEX model; LR+ = 6.40 and LR− = 0.63 for subjective assessment and LR+ = 1.85 with LR− = 0.35 for Simple Rules Risks. The ROMA test had moderate diagnostic performance in discriminating benign from malignant cases. The sensitivity and specificity of this algorithm were 50% and 85%, respectively, with the corresponding LR+ = 3.44 and LR− = 0.58. Of all applied prognostic methods, the ADNEX model had the highest diagnostic accuracy of 76%.

## 4. Discussion

In the present study, we found that the IOTA SR risks and the ADNEX model had higher diagnostic accuracy than tumor markers or the ROMA test. Almost 20% of increased risk results, as indicated by the use of the IOTA group models, were missed by tumor markers alone. However, both models performed poorer than in the general population. Strikingly, a vast proportion of BOTs and small-size early invasive cancers in our studied groups had normal serum tumor marker levels and a low risk of malignancy, as calculated with the ROMA algorithm. We consider that finding important for clinical practice because if only basic ultrasound is performed in a patient with a complex adnexal mass and there are normal levels of both tumor markers, a case of early ovarian cancer, either borderline or invasive malignant, may be easily missed. Our results highlight the importance of an appropriate approach to complex morphology adnexal masses in women regardless of their age, menopausal status or serum tumor marker levels. In cases of such difficult lesions, it may be appropriate to perform an expert transvaginal pelvic exam and/or consult the case with the multispecialty gynecologic oncology team [15,17,22]. 

Regrettably, attempts made by gynecologists, radiologists or other non-expert sonographers to interpret ultrasound results are very often subject to considerable errors. When a sonographic examination of complex adnexal mass is performed by less experienced operators, there is a substantial risk of a false negative result, which means misdiagnosing malignancy and understaging women subsequently found to have presumed early stage cancer. The risk of a false positive classification could also be increased and result in overstaging women without a malignant mass. More difficulties in such cases are encountered with the use of various tumor marker expressions. Approaches based on CA125 single cut-off use were not effective in early stage ovarian malignancy detection [33]. In a study by van Gorp et al., a clear trend could be seen from stage I to stage IV disease, where CA125 and HE4 performed significantly poorer when patients with early and late stages of diseases were compared [34]. Moreover, the addition of HE4 or the ROMA to subjective assessment with or without the Simple Rules in the premenopausal group of women with adnexal masses did not produce any improvement when compared with the IOTA Simple Rules or with subjective assessment alone. These results are in line with our current observations, suggesting that measurements of CA125 or HE4 levels or the ROMA test risk assessment should not be recommended as primary diagnostic tools in early stage malignant ovarian masses. Moreover, in such cases, the HE4/ROMA test used for adnexal mass discrimination may be regarded as an unjustified extra cost.

The vast use of pelvic sonography, computed tomography and magnetic resonance imaging in both symptomatic and asymptomatic women results in the incidental finding of adnexal masses that are commonly encountered at all ages. Because the detection of a suspicious pelvic lesion may result in surgical resection of frequently benign masses, such women may be exposed to personal and economic costs related to unnecessary laparoscopy or laparotomy and ovarian resection [35]. Several practical imaging guidelines, other than those proposed by the IOTA group, have recently been updated to reflect this problem.

Patel-Lippman et al. found that two imaging characterization methods, the Simple Rules developed by the IOTA group and the Society of Radiologists in Ultrasound (SRU) guidelines, in particular, were sensitive for identifying ovarian malignancies. However, the positive predictive value (PPV) was low among women presenting to radiology departments, and masses that had been classified as “indeterminate” turned out to harbor one-third of the total malignancies in the studied group of women [36]. 

Wang et al. recently reviewed three major consensus papers published between 2019 and 2020 [37]. The articles included the SRU consensus update on adnexal cysts, the Ovarian-Adnexal Reporting and Data System (O-RADS) US consensus guideline and the American College of Radiology (ACR) white paper on the management of incidental adnexal findings on MRI or computed tomography. All three papers introduced standardized reporting terminology for adnexal masses that was based on evidence from clinical studies and institutional practice patterns. These guidelines should be applied only to non-pregnant women at average risk for ovarian cancer. Despite small differences in follow-up recommendations based on size thresholds, each of those recommendations had the same general goal, i.e., to limit the number of unnecessary imaging follow-up cases. That would hopefully result in saving the patient’s time, significantly lowering medical care costs and anxiety.

Adnexal cystic–solid masses with one cyst locule and one or more papillations but no other solid components in their inner wall are difficult to classify even by expert examiners [24,38,39]. When selected sonographic tumor features were compared in our studied group of women between benign masses and combined BOTs to early stage MOLs, the latter more frequently expressed one or more papillary projections that were more commonly characterized by irregular contours. Our study revealed that benign adnexal lesions were more frequently non-vascularized. However, the lack of vascularization on color Doppler examination was observed in six borderline tumors and four malignant ovarian lesions. Testa et al. found that borderline tumors and ovarian cancers arising in endometrioid cysts frequently showed a vascularized solid component on ultrasound examination [40]. Moro et al. demonstrated that papillary projections were the most typical ultrasound feature of borderline non-invasive malignant serous ovarian lesions. The presence of solid components but few, if any, papillations was the most representative feature of both low-grade and high-grade invasive serous tumors [41]. Our present results confirm this observation.

Mucama et al. recently found that CA125 is the best available, yet insufficiently sensitive biomarker for the early detection of ovarian cancer [42]. Using pre-diagnostic serum samples, the study identified markers with good discrimination for the lag time of 0–9 months. The authors concluded that the discrimination was low in blood samples collected more than 9 months prior to the diagnosis, and none of the markers showed a major improvement in discrimination when added to CA125. However, a review presented by Srivastava et al. revealed that a combination of at least two biomarkers was more effective than the use of single biomarkers in measures for the diagnosis of early stage malignant ovarian lesions [43].

In their recent meta-analysis, Davenport et al. concluded that the ROMA and ADNEX had a significantly higher sensitivity but also a significantly decreased specificity in premenopausal women [44]. Of all reviewed models, the use of the ADNEX risk assessment had the highest sensitivity but also significantly reduced specificity. Chen et al. found that the performance of the ADNEX model was less effective at distinguishing between BOT and stage I ovarian cancer with an AUC value of 0.61 than at the discrimination of benign and malignant lesions, where the AUCs with and without CA125 were 0.94 and 0.93, respectively [45]. Similarly, He et al. [46] concluded that the ADNEX model was easy to use and excellent in discriminating between benign and stage II–IV malignant ovarian masses. However, it was much less effective at distinguishing between BOTs and stage I OCs and between BOTs and ovarian metastasis, with AUCs of only 0.54 and 0.66, respectively. Interestingly, the inclusion of CA125 into the ADNEX model improved the performance in discriminating between early and late stages of ovarian cancers, but the model still demonstrated problems with differentiating BOTs from stage I OC and BOTs from ovarian metastases. 

Peng et al. recently questioned the value of the ADNEX model in early stage malignant and borderline ovarian tumors. The sensitivity of the test was unsatisfactory for the diagnosis of borderline, stage I and metastatic ovarian tumors in their study group [47]. A retrospective analysis of 85 cases of BOTs conducted by Gaurilcikas et al. revealed that the performance of the ADNEX model based on absolute risk estimation varied from 60.3% of correctly classified cases of BOTs when the selected cut-off was set at 20% probability up to 85.9% of correctly classified cases of BOTs when the cut-off value was set at 3% [48]. They concluded that the calculation based on relative risks or absolute risks with a cut-off value of at least 10% should be used for the ADNEX model in this type of ovarian tumor.

The presented study has important limitations in the interpretation of our results. Firstly, the studied group of complex adnexal masses was relatively small with only 26 BOTs and 20 early stage malignant ovarian masses that were compared to 62 cases of benign adnexal lesions. Secondly, the frequency of malignant tumors was at the upper limit of the published range, partially because of a higher prevalence of various cancers in our selected population. This may have introduced some selection bias. However, following a careful search of our database, we have only included all our cases of complex adnexal masses that fulfilled the strict inclusion criteria.

## 5. Conclusions

The diagnostic value of CA125 and HE4 serum tumor markers and the ROMA algorithm as independent modalities for the discrimination between BOTs and early stage adnexal malignant tumors was limited. Subjective assessment based on ultrasound imaging and the IOTA methods such as the SRRisks and ADNEX model may present superior value over tumor marker assessment in women with complex adnexal masses. The ability of the SA, SRR and ADNEX models to discriminate correctly between benign, borderline and malignant early stage tumors should be evaluated in large-scale prospective studies. 

## Figures and Tables

**Figure 1 diagnostics-13-00885-f001:**
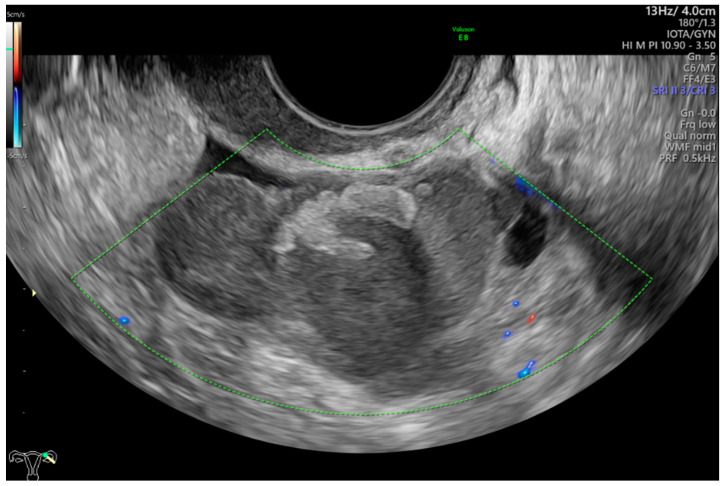
Complex morphology adnexal cyst with mixed echostructure and acoustic shadowing within the cyst. The lack of vascularity indicates IOTA color score of 1. Final histology: benign endometrioma.

**Figure 2 diagnostics-13-00885-f002:**
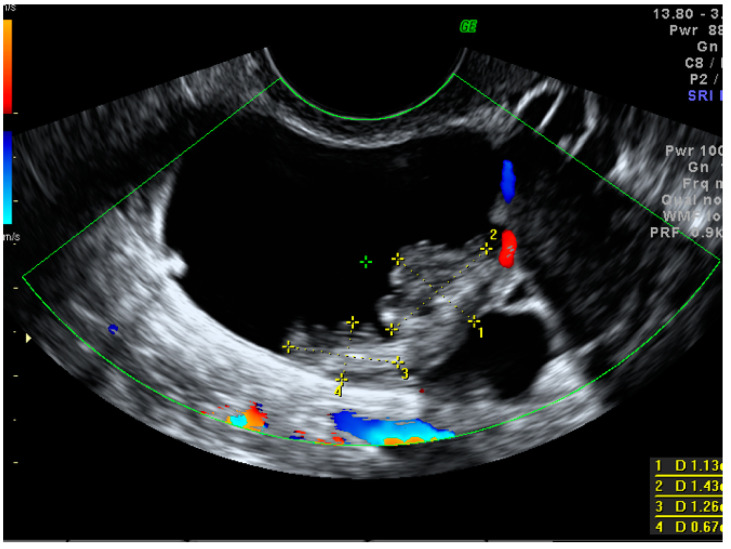
Largest papillary projection measurements in a small borderline ovarian cystic–solid mass.

**Figure 3 diagnostics-13-00885-f003:**
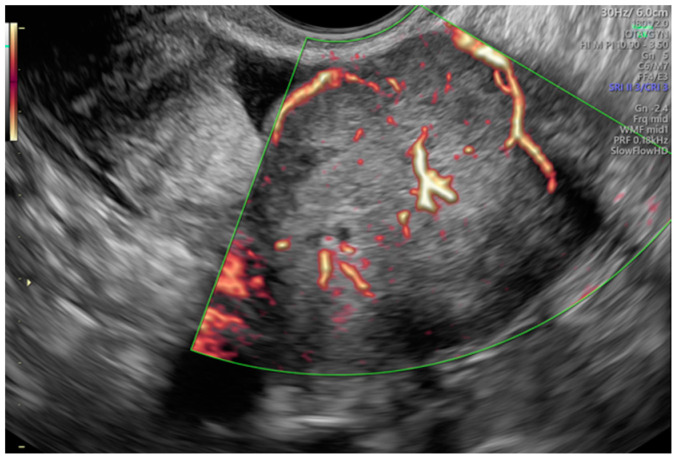
Moderate vascularity of a small solid ovarian mass reflecting the IOTA color score of 3. Final histology: Sertoli–Leydig tumor.

**Figure 4 diagnostics-13-00885-f004:**
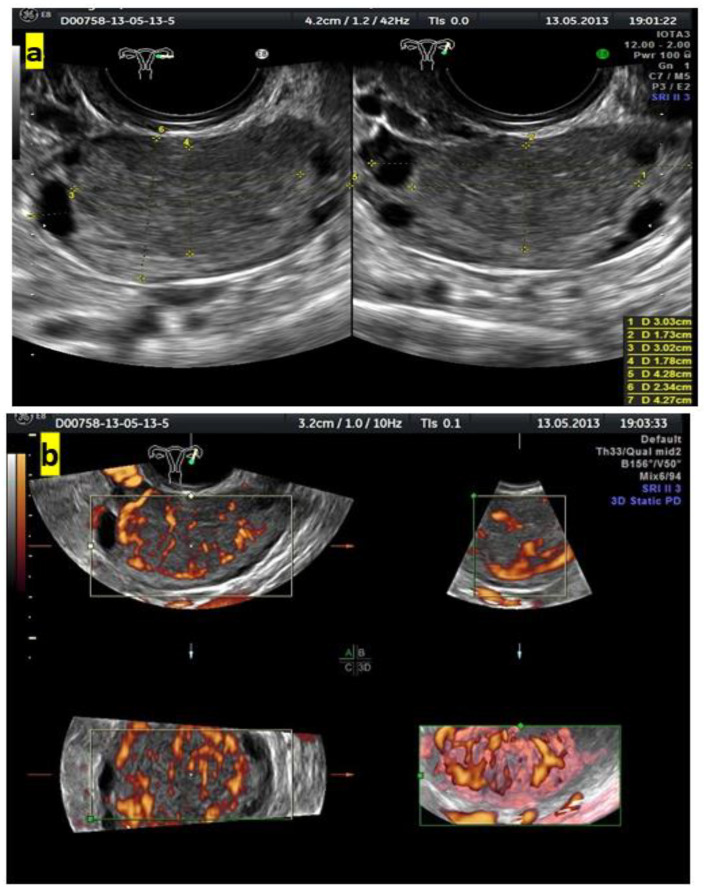
(**a**) Grayscale imaging of two perpendicular sections of a small, oval ovarian lesion resembling “ground-glass” echogenicity. (**b**) Central vascularization (color score, 4) seen in 3 perpendicular planes on color Doppler 3D sonography indicates a solid mass.

**Figure 5 diagnostics-13-00885-f005:**
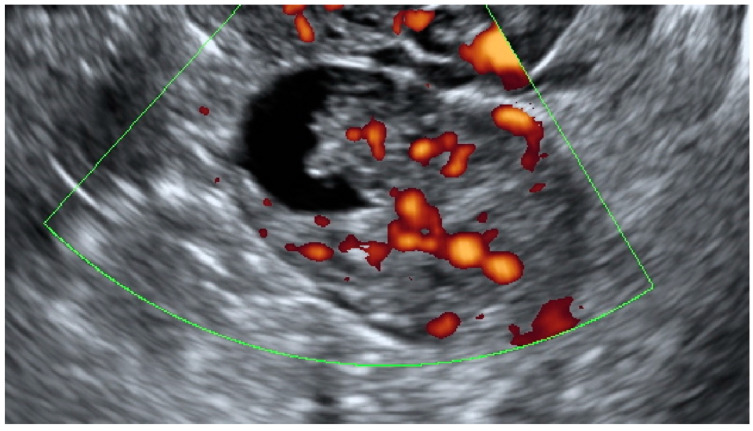
Color Doppler imaging of a small (35 mm in the maximum diameter) cystic–solid lesion with the central vascularization of a papillary projection and a solid part. The final histology revealed early stage invasive serous ovarian cancer.

**Table 1 diagnostics-13-00885-t001:** Histological diagnosis of adnexal tumors in the studied group.

Tumor Histology	(*n* = 108)
Benign tumors	62
Dermoid *	19
Endometrioma	16
Serous cystadenofibroma	12
Mucinous cystadenofibroma	5
Serous cystadenoma	4
Fibroma/fibrothecoma	4
Others **	2
Borderline tumors	26
Serous	16
Mucinous	7
Endometrioid	2
Clear cell	1
Malignant stage I adnexal tumors	20
Serous ovarian cancer	7
Endometrioid ovarian cancer	5
Mucinous ovarian cancer	1
Granulosa cell tumor (adult-type)	5
Oviductal serous cancer	1
Mixed sex cord–stromal tumor	1

* Including two cases of struma ovarii, ** 1 torsion of hemorrhagic cyst and 1 tubo-ovarian abscess.

**Table 2 diagnostics-13-00885-t002:** Age and menopausal status of the studied women with sonographic characteristics of complex adnexal masses.

Age and Menopausal Status	BEN *n* = 62 (57.4%)	BOT *n* = 26 (24.1%)	MOL *n* = 20 (18.5%)	*p*-Value
Age at diagnosis (years) Median (25th–75th percentile) (min–max)	42.5 (30–54) (22–72)	58 (35–66) (16–74)	53.5 (38–60) (16–69)	0.02 BEN vs. BOT 0.03
Postmenopausal status (%)	16 (25.8)	16 (61.5)	12 (60)	0.001
Max diameter (mm) Median (25th–75th percentile) (min–max)	70.5 (50–92) (32–235)	74 (42–124) (23–280)	76 (53.5–113.5) (38–400)	0.46
Max diameter				
<40 mm	4 (6.5)	6 (23.1)	1 (5)	0.002
40–100 mm	48 (77.4)	9 (34.6)	12 (60)	
>100 mm	10 (16.1)	11 (42.3)	7 (35)	
Type of tumor				
Unilocular	12 (19.4)	0 (0)	0 (0)	0.06
Multilocular	8 (12.9)	5 (19.2)	3 (15)	
Unilocular–solid	15 (24.2)	10 (38.5)	5 (25)	
Multilocular–solid	24 (38.7)	9 (34.6)	8 (40)	
Solid	3 (4.8)	2 (7.7)	4 (20)	
Type of echogenicity				
Anechogenic	19 (30.6)	2 (7.7)	7 (35)	0.0002
Low level	15 (24.2)	17 (65.4)	7 (35)	
Ground glass	5 (8.1)	4 (15.4)	4 (20)	
Mixed	12 (19.4)	0 (0)	0 (0)	
Not applicable (solid)	11 (17.7)	3 (10.1)	2 (10)	

BENs—benign masses, BOTs—borderline malignant tumors, MOLs—malignant ovarian lesions, MNPs—postmenopausal patients.

**Table 3 diagnostics-13-00885-t003:** Comparison of selected sonographic features of complex adnexal tumors.

Sonographic Features	BEN *n* = 62 (57.4%)	BOT *n* = 26 (24.1%)	MOL *n* = 20 (18.5%)	*p*-Value
Largest solid component (mm) Median (25th–75th percentile) (min–max)	0 (0–17) (0–102)	19 (0–29) (0–190)	31.5 (9.5–40.5) (0–102)	0.0006 BEN vs. MAL 0.002
Papillary projections				
Present (%)	19 (30.6)	13 (50)	7 (35)	0.22
Number of papillary projections				
0	43 (69.4)	13 (50)	13 (65)	0.01
1	16 (25.8)	4 (15.4)	1 (5)	
2	2 (3.2)	1 (3.9)	2 (10)	
3	0 (0)	3 (11.5)	1 (5)	
4	1 (1.6)	3 (11.5)	1 (5)	
>4	0 (0)	2 (7.7)	2 (10)	
Papillation size Median (25th–75th percentile) (min–max)	0 (0–11) (0–24)	4 (0–17) (0–52)	0 (0–17) (0–35)	0.16
Height of the largest papillary projection (mm) Median (25th–75th percentile) (min–max)	0 (0–8) (0–21)	3 (0–11) (0–45)	0 (0–10) (0–26)	0.21
Crescent sign present	14 (22.6)	3 (11.5)	1 (5)	0.13
Papillation contour				
No papillations	44 (71)	13 (50)	13 (65)	0.008
Smooth	10 (16.1)	1 (3.8)	4 (20)	
Irregular	8 (12.9)	12 (46.2)	3 (15)	
Papillation flow				
Absent	58 (93.6)	21 (80.8)	15 (75)	0.05
Present	4 (6.4)	5 (19.2)	5 (25)	
Acoustic shadows				
Absent	39 (62.9)	21 (80.8)	16 (80)	0.14
Present	23 (37.1)	5 (19.2)	4 (20)	
Microcystic pattern				
Absent	52 (83.9)	20 (76.9)	18 (90)	0.49
Present	10 (16.1)	6 (23.1)	2 (10)	
Color Score				
1	33 (53.2)	6 (23.1)	4 (20)	0.0009
2	13 (21)	13 (50)	4 (20)	
3	15 (24.2)	5 (19.2)	8 (40)	
4	1 (1.6)	2 (7.7)	4 (20)	

BENs—benign masses, BOTs—borderline tumors, MOLs—malignant ovarian lesions, MNPs—postmenopausal patients. The results are presented as medians (range) or numbers and percentages.

**Table 4 diagnostics-13-00885-t004:** CA125 and HE4 concentrations in the studied group of patients.

Tumor Marker Concentrations	BEN *n* = 62 (57.4%)	BOT *n* = 26 (24.1%)	MOL *n* = 20 (18.5%)	*p*-Value
Ca125 (U/mL) Median (25th–75th percentile) (min–max)	21.31 (13.09–35.18) (5–551.9)	18.53 (14.41–45.82) (4.3–627.2)	21.2 (13.7–40.46) (6.23–395.6)	0.77
HE4 Median (25th–75th percentile) (min–max)	48.1 (39.9–60.6) (27.4–120.3)	54.46 (47.9–76) (33.6–147.2)	59.15 (44.95–71.2) (32.6–202.4)	0.09

BENs—benign masses, BOTs—borderline tumors, MOLs—malignant ovarian lesions, MNPs—postmenopausal patients.

**Table 5 diagnostics-13-00885-t005:** Performance of the ROMA, SA, SRR and ADNEX prognostic algorithms in the studied groups.

Test Type	BEN *n* = 62 (57.4%)	BOT *n* = 26 (24.1%)	MOL *n* = 20 (18.5%)	*p*-Value
ROMA Median (25th–75th percentile) (min–max)	7.9 (5.2–13.5) (1.9–42.1)	13.45 (9.49–24.9) (3.4–80.2)	13.95 (7.75–21.25) (2.9–51)	0.001 BEN vs. BOT 0.003 BEN vs. MAL 0.047
ROMA class				
Low risk	53 (85.5)	16 (61.5)	7 (35)	<0.001
High risk	9 (14.5)	10 (38.5)	13 (65)	
SRR Median (25th–75th percentile) (min-max)	15.2 (1.1–27.5) (0.1–81.4)	38.1 (27.5–71.7) (1.1–98.4)	48.7 (27.5–76.55) (0.9–81.7)	<0.001 BEN vs. BOT <0.001 BEN vs. MAL <0.001
SRR risk of malignancy stratification				
Low risk	2 (3.2)	2 (7.7)	0 (0)	<0.001
Intermediate risk	33 (53.2)	4 (15.4)	3 (15)	
High risk	27 (43.6)	20 (76.9)	17 (85)	
ADNEX risk (%) Median (25th–75th percentile) (min–max)	5.2 (0.8–16.9) (0.3–58.3)	37.25 (5.7–72.8) (1.6–97.6)	33.95 (18.65–52.8) (2.9–82.1)	<0.001 BEN vs. BOT <0.001 BOT vs. MAL <0.001
ADNEX risk of malignancy				
Low risk	22 (35.5)	3 (11.5)	1 (5)	<0.001
Intermediate risk	28 (45.2)	6 (23.1)	4 (20)	
High risk	12 (19.3)	17 (65.4)	15 (75)	
ADNEX Risk				
BEN	50 (80.7)	9 (34.6)	5 (25)	<0.001
MAL	12 (19.3)	17 (65.4)	15 (75)	
SA risk of malignancy				
BEN	47 (75.8)	8 (30.8)	4 (20)	<0.001
BOT	11 (17.7)	10 (38.4)	5 (25)	
MAL	4 (6.5)	8 (30.8)	11 (55)	

BENs—benign masses, BOTs—borderline tumors, MOLs—malignant ovarian lesions. The results are presented as medians (range) or numbers and percentages. SRRs—Simple Rules Risks, SA—subjective assessment.

**Table 6 diagnostics-13-00885-t006:** Predictive values of the ROMA, ADNEX, SA and SRR in the group of 62 women with benign masses compared to 46 patients with borderline or early stage malignant adnexal lesions.

Index Test	TP	FP	FN	TN	SENS	SPEC	LR+	LR−	PPV	NPV	ACC	Chi^2^	*p* Value
ROMA	23	9	23	53	0.50	0.85	3.44	0.58	0.72	0.70	0.70	chi^2^ = 15.9	*p* < 0.0001
ADNEX	32	12	14	50	0.70	0.81	3.59	0.38	0.73	0.78	0.76	chi^2^ = 27.6	*p* < 0.0001
SA	19	4	27	58	0.41	0.94	6.40	0.63	0.83	0.68	0.71	chi^2^ = 17.1	*p* < 0.0001
SRR	37	27	9	35	0.80	0.56	1.85	0.35	0.58	0.80	067	chi^2^ = 14.9	*p* = 0.0001

SA—subjective assessment for borderline and invasive malignant tumors; SRRs (1 + 2)—Simple Rules Risks for borderline and malignant invasive tumors; TP—true positive; FP—false positive; FN—false negative; TN—true negative; SENS—sensitivity; SPEC—specificity; LR+—positive likelihood ratio; LR−—negative likelihood ratio; PPV—positive predictive value; NPV—negative predictive value; ACC—accuracy.

## Data Availability

The data set is available in Appendix A.

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
