# Peer review of "Performance of IOTA Simple Rules Risks, ADNEX Model, Subjective Assessment Compared to CA125 and HE4 with ROMA Algorithm in Discriminating between Benign, Borderline and Stage I Malignant Adnexal Lesions"

_diagnostics, 2023, doi:10.3390/diagnostics13050885_

Round 1

Reviewer 1 Report

The authors have performed this study with a scientific rigor. 

Hoever the following points to be considered and incorporated into the manuscript:

1. The conceptual frame work is missing in the manuscript that is critical to this manuscript.

2. More recent references have to be included and discussed with the findings obtained in this study inorder to strengthen the discussion section of the manuscript.

Author Response

Reviewer #1

Query: “The authors have performed this study with a scientific rigor. However the following points to be considered and incorporated into the manuscript: 1. The conceptual frame work is missing in the manuscript that is critical to this manuscript.”

Response: We are very thankful for such a positive review. The aim of our study was to describe currently used scoring systems based on ultrasound examination of adnexal masses in women and to compare these methods with the measurements of two popular ovarian tumor markers serum expression and their combination called the ROMA algorithm. We are sorry that in the first version of the manuscript we were not able to show that conceptual framework explicitly. However, we think that this can be still partially seen in the Abstract, where several parts of the framework can be inferred and they are listed under: “Background”, “Objective”, “Conclusions” of this part. According to the relevant literature, the structure of a theoretical framework is fluid, and there are no specific rules that need to be followed, as long as it is clearly and logically presented. We have chosen a descriptive rather instead of a graphical type of framework. In the Introduction section we have included several important components of the framework, namely a literature review of recent studies associated with our thesis topic as well as theories/models used in the studied field of adnexal masses imaging research.

The challenge in determining the isolated potential benefit from US and tumor markers strategy is that in large clinical studies that provided rich data, scoring systems are often analyzed together for various types of masses, making it difficult to ascertain differences between these subsets of several types of complex benign, low malignant potential lesions (BOTs) and early stage invasive malignant ovarian masses. Furthermore, to date only few studies have directly examined BOTs, early ovarian cancers against tumor markers/expert opinion in the context of ovarian neoplasms, as both methods have been shown to be less effective in discrimination between various types of benign and early malignant lesions. To be effective in detecting ovarian BOTs and early malignant cases we have proposed an approach based on comparisons of various currently used imaging and biochemical methods. Since it is crucial to identify other variables that can influence the relationship between our independent and dependent variables we have included moderating, mediating, and control variables such as: patient’s age and menopausal status, CA125 and HE4 serum concentrations, tumor size and lesion’s echogenic structure. We have also analyzed multiple tumor features found on ultrasound examination. They included: tumor type, cyst echogenicity, three various tumor sizes groups, presence of ovarian crescent sign, presence of acoustic shadows and presence of solid parts and papillary projections. In order to  test a cause-and-effect relationship, we have identified two key variables: independent and dependent variables. In this particular study the expected cause, tumor ultrasound morphology and the expression of selected tumors markers were “the predictor”, i.e. independent or explanatory variables, whereas the expected effect, “risk score or probability,” was the “dependent”, i.e. the response, or outcome variable.

The literature review acted as a filtering tool to select appropriate thesis questions and to guide our data collection followed by a specific data analysis, and the interpretation of our findings. Such a broad interpretation can be found in a Discussion section. Apart from reviewing relevant published papers in our field of research, we have also explored theories that we had to come across in other published studies and textbooks. One such theory is for example the suggestion to use of ROMA algorithm in discrimination of adnexal masses and clinical decision making that could triage women for surgery. We have shown that both CA125 and HE4 assessment alone or as ROMA algorithm had little, if no value at all in the discrimination of complex morphology adnexal masses.

Therefore, the conclusion reached in this manuscript assumes that in our cohort study of a selected patient population, the SRR and ADNEX risk stratification systems performed within the expected range as previously published in the literature. We  have also added the important limitations in the interpretation of our results. The relevant text starts in line 513 and goes like this: ”The presented study has important limitations in the interpretation of our results. Firstly, the studied group of complex adnexal masses was relatively small with only 26 BOTs and 20 early stage malignant ovarian masses that were compared to 62 cases of benign adnexal lesions. Secondly, the frequency of malignant tumors was at the higher end of the published range, partially because of the higher prevalence of various cancers in our selected population. This may have introduced some selection bias, however, following a careful search of our database, we have included all consecutive cases of complex masses that fulfilled the strict inclusion criteria.”

Query: “2. More recent references have to be included and discussed with the findings obtained in this study in order to strengthen the discussion section of the manuscript.”

Response: We agree with the Reviewer’s suggestion that the Discussion section of the manuscript could be improved by the addition of several most recent references.

We have added 5 new citations and text starting in the following lines:

Line 410: “Prevailing methods for early diagnosis of ovarian malignant masses currently include ultrasound transvaginal imaging, several serum biological marker examination, or a blend of the imaging techniques with selected biochemical methods [35]. The vast use of pelvic sonography, computed tomography, and magnetic resonance imaging in both symptomatic and asymptomatic women results in incidental findings of adnexal masses detection that are commonly encountered at all ages. Since the detection of the suspicious pelvic lesion may result in surgical resection of mostly benign masses, these women may be exposed to personal and economic costs related to unnecessary laparoscopy or laparotomy and ovarian resection. Thus, accurate non-invasive risk of malignancy stratification in detected adnexal masses is essential for optimal patient’s management.

Patel-Lippman  et al. have found that two imaging characterization methods, in particular the Simple Rules developed by the IOTA group and SRU guidelines were sensitive for identifying ovarian malignancies, but the positive predictive value (PPV) was low among women presenting to radiology departments, and masses that were classified as “indeterminate” turned out to harbor one-third of the total malignancies of the studied group of women [36]. On the other hand there is a compelling evidence that at pelvic imaging the vast majority of incidental cystic adnexal masses found in women are almost always benign. Several other than the proposed by the IOTA group practical imaging guidelines were recently updated to reflect this problem.  Wang et al. have reviewed three major consensus papers published between 2019 and 2020. These articles included the Society of Radiologists in Ultrasound (SRU) consensus update on adnexal cysts, the Ovarian-Adnexal Reporting and Data System (O-RADS) US consensus guideline, and the American College of Radiology (ACR) white paper on the management for incidental adnexal findings at MRI or computed tomography [37].  All three papers introduced standardized reporting terminology for adnexal masses that was based on evidence from clinical studies and institutional practice patterns. These guidelines should be applied only in non-pregnant women of average risk for ovarian cancer. Despite small differences in follow-up recommendations based on size thresholds each of these recommendations had the same goal, that generally was to limit the number of unnecessary imaging follow-up cases.

Ultrasonography-based morphology scoring systems can be used to differentiate benign from malignant adnexal masses. These scoring systems are based on specific ultrasound parameters, each with several scores base on determined features. All evaluated scoring systems were found to have an acceptable level of sensitivity and specificity; the choice of scoring system may therefore be made based on clinician preference. As a standalone modality, serum cancer antigen 125 is not recommended for distinguishing between benign and malignant adnexal masses.

Line 463: Mucama et al. have found recently that CA125 is the best available yet insufficiently sensitive biomarker for early detection of ovarian cancer [42]. There is a need to identify novel biomarkers, which individually or in combination with CA125 can achieve adequate sensitivity and specificity for the detection of earlier-stage ovarian cancer. Using pre-diagnostic serum samples, this study identified markers with good discrimination for the lag-time of 0-9 months. The Authors have concluded that the discrimination was low in blood samples collected more than 9 months prior to diagnosis, and none of the markers showed major improvement in discrimination when added to CA125. However, the review presented by Srivastava et al. revealed that a combination of at least two biomarkers had beaten single biomarkers in measures for the diagnosis of the early stage malignant ovarian lesions [43].

Reviewer 2 Report

Original article: “Performance of IOTA Simple Rules Risks, ADNEX model, sub- 2 jective assessment compared to CA125 and HE4 with ROMA al- 3 gorithm in discriminating between benign, borderline and 4 stage I malignant adnexal lesions“ presents different diagnostic modes in ovarian tumors according to modern postulates. The materials and methods are well specified, and the results are clearly presented. The discussion is adequate with references accompanying the discussion. For the special issue of the magazine, this paper can contribute to this issue and the topic of ovarian tumorous masses.

Author Response

Reviewer #2: Author's response.

We are extremely grateful for a positive review of our manuscript. In an answer to the suggestion of possible conclusions section improvement we have added some comments on the limitations in the interpretation of our results that we think would fit into this category. Specifically, at the end of the Discussion section of our revised manuscript we have added the following text:  

”The presented study has important limitations in the interpretation of our results. Firstly, the studied group of complex adnexal masses was relatively small with only 26 BOTs and 20 early stage malignant ovarian masses that were compared to 62 cases of benign adnexal lesions. Secondly, the frequency of malignant tumors was at the higher end of the published range, partially because of the higher prevalence of various cancers in our selected population. This may have introduced some selection bias, however, following a careful search of our database, we have included all consecutive cases of complex masses that fulfilled the strict inclusion criteria.”

We hope that this answer will satisfy the Reviewer.